# Impact of the COVID-19 pandemic on the mental health of professionals in 77 hospitals in France

Alicia Fournier[1]*, Alexandra Laurent[1,2], Florent Lheureux[3], Marie Adèle Ribeiro-Marthoud[4], Fiona Ecarnot[5,6], Christine Binquet[7,8], Jean-Pierre Quenot[9,10,11,12]

**1** Laboratoire de Psychologie: Dynamiques Relationnelles Et Processus Identitaires (PsyDREPI), Université de Bourgogne Franche-Comté, Dijon, France, **2** Département d'anesthésie et médecine chirurgicale, CHU François Mitterrand, Dijon, France, **3** Psychology Laboratory, University of Burgundy Franche-Comté, Besançon, France, **4** Direction des soins, Centre Hospitalier Universitaire Dijon-Bourgogne, Dijon, France, **5** Department of Cardiology, University Hospital Besançon, Besançon, France, **6** EA3920, University of Burgundy Franche-Comté, Besançon, France, **7** Inserm, CIC1432, module Epidémiologie Clinique, Dijon, France, **8** CHU Dijon-Bourgogne, Centre d'Investigation Clinique- Epidémiologie Clinique/Essais Cliniques, Dijon, France, **9** CHU Dijon-Bourgogne, Service de Médecine Intensive-Réanimation, Dijon, France, **10** Equipe Lipness, centre de recherche INSERM, Université de Bourgogne Franche-Comté, UMR1231 Lipides Nutrition Cancer, Lipness, Dijon, France, **11** FCS Bourgogne-Franche Comté, LipSTIC LabEx, Dijon, France, **12** Espace de Réflexion Éthique Bourgogne Franche-Comté (EREBFC), Dijon, France

* alicia.fournier@u-bourgogne.fr

**Data Availability Statement:** The data is available on: https://osf.io/yxqvg/.

**Funding:** Supported in part by funds from J-P. Quenot provided through the French Ministry for

## Abstract

The COVID-19 pandemic has led to significant re-organisation of healthcare delivery in hospitals, with repercussions on all professionals working in healthcare. We aimed to assess the impact of the pandemic on the mental health of professionals working in health care institutions and to identify individual and environmental factors influencing the risk of mental health disorders. From 4 June to 22 September 2020, a total of 4370 professionals responded to an online questionnaire evaluating psychological distress, severity of post-traumatic stress symptoms, stress factors, and coping strategies. About 57% of the professionals suffered from psychological distress, and 21% showed symptoms of potential post-traumatic stress. Professionals working in radiology, those working in quality/hygiene/security and nurses' aides were the most affected groups. The media focus on the crisis, and a high workload were the most prevalent stress factors, followed by uncertainty regarding the possibility of containing the epidemic, the constantly changing hygiene recommendations/protocols, and the lack of personal protective equipment. The use of coping strategies, notably positive thinking, helped to mitigate the relation between perceived stress and mental health disorders. The COVID-19 pandemic has had far-reaching negative repercussions for all professionals, with some sectors more markedly affected. To prevent mental health disorders in professionals during a public health crisis, support services and management strategies within hospitals should take account of the importance of positive thinking and social support.

Health, who partially funded this study through the national programme for hospital research (Programme Hospitalier de Recherche Clinique National, PHRC-COVID 2019). The funders had no role in study design, data collection and analysis, decision to publish, or preparation of the manuscript. The authors thank the French Ministry for Health, who funded this study through the national programme for hospital research (Programme Hospitalier de Recherche Clinique National, PHRC-COVID 2019). There was no additional external funding received for this study.

**Competing interests:** The authors have declared that no competing interests exist.

## Introduction

During the first wave of the COVID-19 pandemic in France, there was a profound re-organisation of healthcare delivery in hospitals in France in order to cope with the massive influx of patients. These measures included organisational restructuring (such as changing admission circuits for patients and their management trajectory within the hospital), hygiene measures (separating COVID-19 from non-COVID-19 wards), and logistics (transport of materials and patients). Interpersonal relations were also affected, with reduced contact between staff (forbidden to gather for coffee/meal breaks etc), and between teams from different wards or departments (to avoid possible clusters of COVID-19 infections). In addition to these far-reaching, and sudden changes to the work organisation, many healthcare workers lacked adequate personal protective equipment (PPE) and many units were under-staffed, especially in the zones that were hardest hit at the beginning of the first wave, jeopardizing optimal management of patients and optimal protection of healthcare workers from possible contamination [1,2]. On top of these difficulties, there was also a strong feeling of personal insecurity, due to the lockdown, the risk of contaminating families [1] and the uncertainty regarding the outcome of the pandemic [1,3,4].

Numerous studies have investigated the impact of the COVID-19 pandemic on healthcare workers in units dedicated to the care of COVID-19 patients, reporting high levels of anxiety, depression, burnout, insomnia and psychological distress [5–10]. Among health professionals, meta-analyses reveal a frequency of 12.2–36% for depression and 13–37% for anxiety [11–14]. Also, a recent review and meta-analysis conducted in October 2020 among health care workers revealed a frequency of 30.0% of anxiety, 31.1% of depression, 56.5% of acute stress, 20.2% of potential post-traumatic stress and 44.0% of sleep disorders [15]. However, all healthcare workers did not develop mental health issues directly due to the pandemic. This could be explained by the fact that the negative impact of stress on mental health can be modulated by the use of coping strategies [16]. A positive attitude towards the pandemic could be protective, while seeking social support and avoidance strategies are reported to be factors that may compound the risk of psychological distress [17]. While the consequences of the pandemic for healthcare workers in COVID-19 units are now well established, data remain sparse regarding other professionals working in other hospital departments and services. Yet, new recommendations about working conditions in healthcare establishments [18,19] underline that workers across all professions have been affected, and not only those working directly with COVID-19 patients.

In this context, using a mixed methods approach (quantitative and qualitative), the PsyCO-VID–All Professionals study [Psychological support for health care professionals in hospital in the COVID-19 pandemic context]) aimed to assess the frequency of psychological distress, and its impact on professionals across all sectors, and to identify sources of stress related to the COVID-19 pandemic. Secondary objectives were to identify factors at individual and environmental level related to the risk of developing mental health disorders (psychological distress, and psycho-traumatic impact), while taking account of coping strategies as mediators of this relation.

## Materials and methods

### Study design

We performed a cross-sectional, multicentre study in 73 Departments in France (Fig 1 and S1 Table), from 4 June to 22 September 2020, using an online questionnaire distributed via the Limesurvey platform.

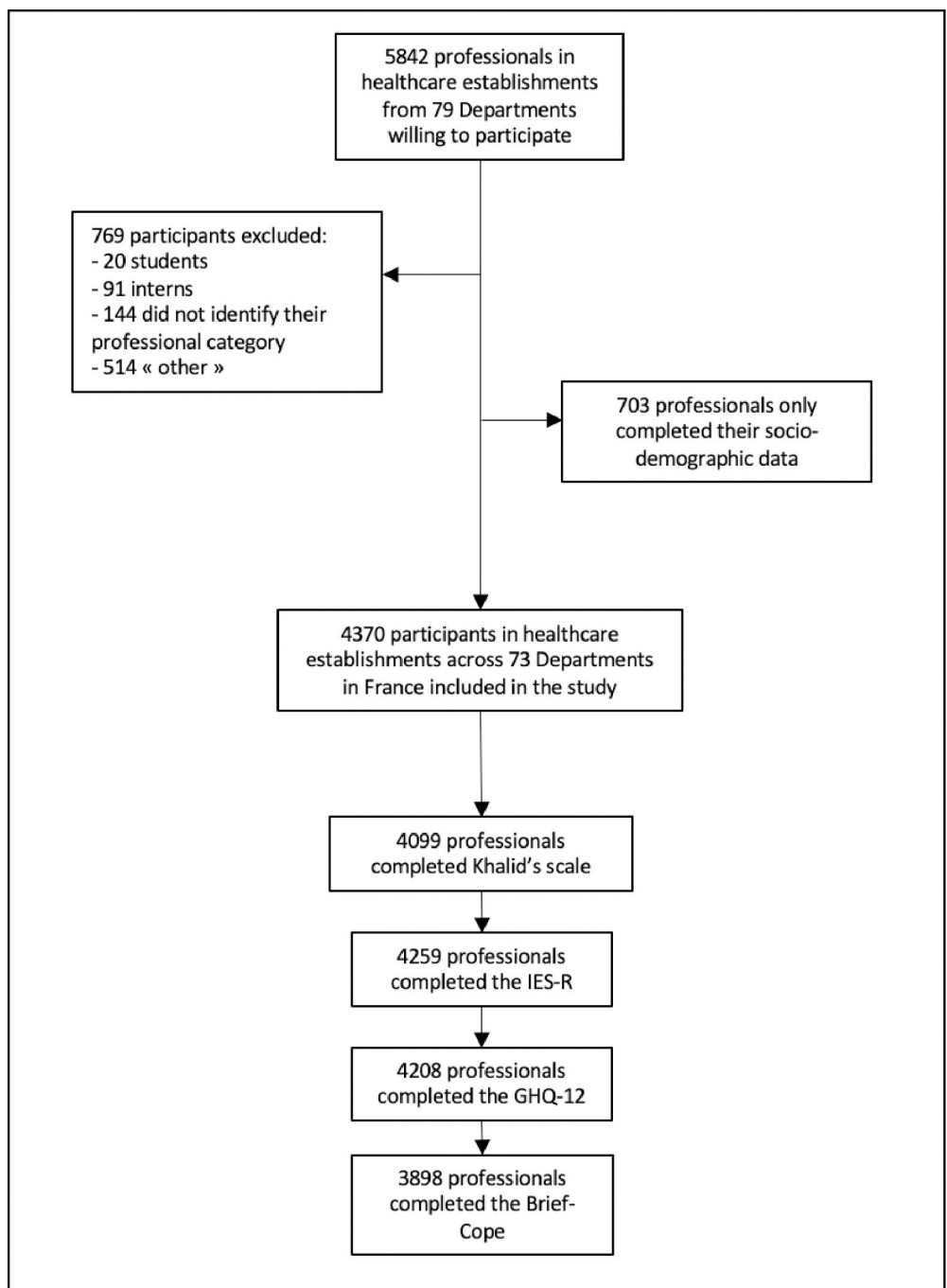

**Fig 1. Design of the cross-sectional, multicentre Psy-COVID-All professionals study, performed from 4 June to 22 September 2020.**

## Study population

All permanent or contractual professionals working in medical/caregiving professions (nurses' aides, nurses, doctors, social workers, biologists/laboratory technicians, pharmacists/pharmacy technicians, psychologists, nursing managers, physiotherapists, midwives, clinical research staff) or in non-medical professions (welcome desk, admissions office, administration,

logistics/procurement, quality/hygiene/security/environment (QHSE)), instructors, maintenance workers, computer engineers) in public or private hospitals who were participating in the PsyCOVID study (NCT04357769) were eligible. Non-inclusion criteria include students of the medical and paramedical professions, and interns. In a second stage, we excluded professionals who did not clearly indicate their professional category, and professions for which there were fewer than 50 completed questionnaires (Fig 1).

## Study implementation

All professionals were informed about the study objectives and procedures, and were given the link to participate in the study by their hospital administration. We also relayed information about the study via the internal communication channels within each participating hospital (intranet, newsletter etc), and orally via the department chiefs, and in written format via posters in common areas. Participating professionals were required to accept the terms of the study before responding to the questionnaire. The responses were confidential and anonymous. The study was registered at ClinicalTrials.gov: NCT04944394. The study was approved by the Ethics Committee of the French Intensive Care Society (N˚20–33) on 21 April 2020.

## Endpoints

**Primary endpoint.** Mental health among professionals was evaluated using the validated French version of the 12-item General Health Questionnaire (GHQ-12) [20]. The GHQ-12 is a self-report measure of the severity of psychological morbidity in non-psychiatric settings, and measures change in mental state following upsetting events, by assessing symptoms related to psychological distress and general functioning. We used the standard scoring method (0–0–1–1), which gives a possible score ranging from 0 to 12, whereby a higher score indicates a greater degree of psychological distress. A threshold of 3 or more (2/3) has been used to identify the presence of psychological distress in other studies [21–24].

**Secondary outcomes.** To assess the psycho-traumatic impact of the pandemic, we measured symptoms of post-traumatic stress disorder (PTSD) using the Impact of Event Scale-Revised (IES-R) in its validated French version [25,26]. The IES-R is a self-report scale evaluating the severity of PSTD symptoms after stressful life events, and respondents report their level of difficulty over the previous 7 days. The 22 items are rated on a Likert scale ranging from 0 (not at all) to 4 (extremely). The scale comprises 3 dimensions (avoidance, intrusion and hyperarousal) whose scores are obtained by averaging the scores of the items of that dimension. The total IES-R score ranges from 0 to 88, and at more than one month after a traumatic event, a score >33 signifies the likely presence of possible PTSD [27–29].

To measure sources of stress and the intensity of the stress perceived, we used items from the scale developed by Lee et al [30] during the severe acute respiratory syndrome (SARS) epidemic in Taiwan in 2003, and adapted by Khalid et al [31] during the 2015 MERS-CoV outbreak in Saudi Arabia. This scale comprises 5 sections, namely: exploration of the emotions experienced, identification of perceived stress factors and their intensity, availability of resources within the hospital to help professionals copy; coping strategies used by the professionals, and motivating factors to participate in a future epidemic. To meet the study's objectives of measuring perceived stress, we retained 13 items that we adapted to the COVID-19 pandemic situation, from the section relating to perceived stress factors and their intensity, corresponding to the items most frequently reported by professionals and that were most strongly associated with stress in the study by Khalid et al. The items were evaluated on a 5-point Likert scale ranging from 0 (I did not experience this situation) to 4 (and I was very much stressed"). The scores of the 13 items were summed and averaged, yielding an overall mean perceived stress score ranging from 0 to 4.

Coping strategies were assessed using the Brief-COPE questionnaire [32,33]. Four types of coping were assessed (social support seeking, problem solving, avoidance and positive thinking) that are likely to act as a buffer against stressful events [34,35]. Higher scores reflect a greater tendency to implement the corresponding coping strategy.

Finally, participants were asked whether they had experienced any stressful life events since the beginning of the epidemic, either related to COVID-19 (e.g., had symptoms of or was diagnosed with COVID-19, had a family member who had symptoms of or was diagnosed with COVID-19, had a family member who died of COVID-19), or other difficult life events unrelated to the epidemic.

In addition, via an open question at the end of the questionnaire, respondents were given the opportunity to describe a maximum of 10 situations related to their profession that they had found particularly stressful during the epidemic.

## Data analysis

**Analysis of data from clinical scales.** Quantitative variables are described as mean±standard deviation (SD) and categorical variables as number (percentage). We describe the scores obtained on the IES-R, GHQ-12 and perceived stress scale according to the type of profession. To compare medical vs non-medical staff, we used ANOVA or Welch's F test, as appropriate.

To identify factors associated with the severity of PTSD symptoms and with the severity of psychological distress, we used descending hierarchical linear models using the lm function in R (R lme4 [36,37]). Since the impact of the sources of stress on psychological distress depends on the use of specific coping strategies [34], we evaluated whether coping strategies (as assessed by the Brief-COPE) mediated the association between perceived stress during the epidemic, and the severity of psychological distress (as assessed by the GHQ-12). We adopted the same approach for the association between intensity of perceived stress during the epidemic, and the severity of PTSD symptoms (as assessed by the IES-R). For individual and contextual variables (e.g., sex, age, living conditions, marital status, changes to work schedule, number of hours worked, changes to working hours, having experienced a difficult life event related to COVID-19, and job title) that could affect the relation between stress and mental health, we first tested a full model including all variables, to identify those that were associated with mental health. Then, we progressively tested new models including only variables that were significant in the first step. Finally, we used the anova function in R lme4 [36,37] to compare models and identify whether removal of any variables would significantly improve model fit. The model with the best fit was chosen according the Akaike Information Criterion (AIC) [38].

All analyses were performed using R (version 1.3.959) and SPSS (version 26) for Macintosh. A $p$ value <0.05 was considered statistically significant.

**Analysis of the open-ended questions.** We analysed the data yielded by the open-ended questions according to the procedure described by Clarke et al [39]. All responses were read in detail and coded. Themes were identified by two researchers (AF, AL) for each profession. Themes were discussed until consensus was reached with a third researcher (FL). The main themes that emerged are described in table format. Analyses were performed with the aid of QSR International's NVivo 10 qualitative data analysis software.

## Results

### Socio-demographic characteristics

A total of 4370 professionals from 77 hospitals across 73 Departments of France were included (S1 Table). The majority of respondents were women (n = 3570, 81.7%) and either married or living maritally (n = 3367, 77%). Among the medical/caregiving staff, (n = 3203, 73.3%), 919

(28.7%) were nurses and 730 (22.8%) were physicians; while among the non-medical staff (n = 1167, 26.7%), 520 (44.6%) had administrative position, and 232 (19.9%) worked on welcome desks (Table 1).

## Impact of the epidemic on frequency and severity of psychological distress

Using the cut-off value for the GHQ-12 (GHQ-12≥3), and considering all professions, a total of 56.9% of professionals presented psychological distress (56.7% among the medical/caregiving staff *vs.* 57.4% among the non-medical staff, *p* = .654). Midwives were the most affected (69.4%), followed by professionals working in QHSE (67.7%), while psychologists and those working in technical maintenance/computer networks had the lowest levels (respectively 46.9% and 46.9%).

The mean GHQ-12 scores by profession are shown in Fig 2. The mean GHQ-12 score overall was 3.8±3.1. Comparisons between medical and non-medical staff did not show any significant difference in the average psychological distress scores (medical/caregiving staff 3.8±3 *vs.* non-medical staff 3.7±3.1).

## Impact of the epidemic on the frequency and severity of PTSD symptoms

Considering all professions, a total of 21.2% professionals suffered from possible PTSD (IES-R>33), namely 21.2% among medical/caregiving staff *vs.* 21.4% among non-medical staff, *p* = .892); and 19.4% and 20.7% of medical and non-medical staff respectively presented both potential PTSD and psychological distress. There was no difference between the main occupational categories (*p* = .358).

Professionals working in radiology (36.4%) were most strongly affected by PTSD, followed by nurses' aides (35.5%) and professionals working in the QHSE sector (35.1%). Psychologists were least affected (10.8%). Furthermore, professionals in radiology (33.9%), QHSE (33.1%) and nurses' aides 32.5%) were those that had the highest occurrence of both potential PTSD and psychological distress (S2 Table).

Average scores on the IES-R are show in Fig 3 by profession. Overall, the mean score was 20.4±18.1. The psycho-traumatic impact was most marked in the intrusion dimension (1.1±1) compared to the two other dimensions (all *p*<0.001). Comparisons between medical and non-medical personnel did not show any significant difference in IES-R scores (medical/caregiving staff 20.4±18.1 *vs.* non-medical 20.6±18.2).

## Intensity of perceived stress since the start of the pandemic, and stress factors

**Intensity of perceived stress.**  The average scores on Khalid's stress scale are show in Fig 4, by profession. The most COVID-19-related stress factor with the highest impact, and common to all professions, was media coverage of the COVID-19 crisis. The item "Not knowing when the epidemic would be brought under control" also scored highly in the majority of professions (i.e., nurses' aides, laboratory staff, pharmacy staff, radiology staff, psychologists, physiotherapists, midwives, physicians, maintenance staff, procurement/logistics, QHSE, instructors). The item "Recommendations and protocols are constantly changing" presented high scores for staff working in laboratories, psychologists, nursing managers, physiotherapists, radiology staff, midwives, QSHE staff and instructors.

In general, medical/caregiving staff had higher stress scores related to COVID-19 than non-medical staff (1.8±0.7 *vs.* 1.4±0.7 respectively, *p*<0.001). The professional groups with the most COVID-19-related stress were nurses' aides (2.1±0.8), nurses (2±0.8) and radiology staff

**Table 1. Socio-demographic characteristics of the study population—PsyCOVID all professionals study, performed from 4 June to 22 September 2020.**

| | | Total |
|---|---|---|
| **Number** | | 4370 (100) |
| **Sex** | | |
| | Females | 3570 (81.7) |
| | Males | 800 (18.3) |
| **Age, years** | | |
| | 18–29 | 467 (10.7) |
| | 30–44 | 1950 (44.6) |
| | 45–60 | 1812 (41.5) |
| | > 60 | 141 (3.2) |
| **Profession** | | |
| | **Medical/caregiving** | |
| | Nurses' aides | 315 (7.2) |
| | Nurses | 919 (21) |
| | Physicians | 730 (16.7) |
| | Working in a laboratory | 118 (2.7) |
| | Working in the pharmacy | 114 (2.6) |
| | Psychologists | 199 (4.6) |
| | Nursing managers | 396 (9.1) |
| | Physiotherapists | 61 (1.4) |
| | Working in radiology | 67 (1.5) |
| | Social workers | 81 (1.9) |
| | Midwives | 74 (1.7) |
| | Clinical research staff | 129 (3) |
| | **Non-medical professions** | |
| | Welcome desk/orientation of visitors | 232 (5.3) |
| | Quality/hygiene/security/environment (QHSE) | 136 (3.1) |
| | Administration | 520 (11.9) |
| | Logistics/procurement | 107 (2.4) |
| | Instructors | 58 (1.3) |
| | Technical maintenance and computers/networks | 114 (2.6) |
| **Marital status** | | |
| | Single/divorced/separated/widowed | 949 (21.7) |
| | Married/living maritally | 3367 (77) |
| | Missing data | 54 (1.2) |
| **Change in volume of work compared to normal conditions** | | |
| | Worked less | 207 (4.7) |
| | No change | 2583 (59.1) |
| | Worked more | 1411 (32.3) |
| | Missing data | 169 (3.9) |
| **Change in practical organisation of work** | | |
| | No | 1703 (39) |
| | Yes | 2517 (57.6) |
| | Missing data | 150 (3.4) |
| **Change in living conditions** | | |
| | No | 4158 (95.1) |
| | Yes | 212 (4.9) |

(*Continued*)

**Table 1.** (Continued)

| | | Total |
|---|---|---|
| **Full-time or part-time work** | | |
| | Part time | 850 (19.5) |
| | Full time | 3490 (79.9) |
| | Missing data | 30 (0.7) |
| **Experienced a stressful life event related COVID-19** | | |
| | No | 2277 (52.1) |
| | Yes | 2080 (47.6) |
| | Missing data | 13 (0.3) |

n (%). Change in living conditions = any change between the usual condition before the epidemic ("I live with my family", "I live alone", "other"), and condition during the epidemic.

(2.1±0.7); whereas those with the lowest perceived stress were staff working in instruction/training (1.3±0.6), clinical research (1.3±0.7) and procurement/logistics (1.4±0.7) (Fig 4).

**Stress factors.** Analysis of the answers to the open-ended questions revealed that the workload, the lack of PPE, and the constraints of the changing hygiene protocols were the most commonly cited difficulties, across all professional groups (Table 2). In 14 out of 18 professional groups, we noted indications of excess workload, and in 10 professional groups, reports of a lack of PPE. In addition, working from home, and managing emotions related to colleagues/staff were other frequently cited difficulties. It is noteworthy that among the difficulties cited above, only those working in the QHSE sector actually experienced aggressiveness at the hands of other professions (Table 2). For example, some participants working in this sector cited difficulties such as the aggressiveness of the medical staff towards them, and the impression of being on trial in front of aggressive, not to say violent people. They also cited the

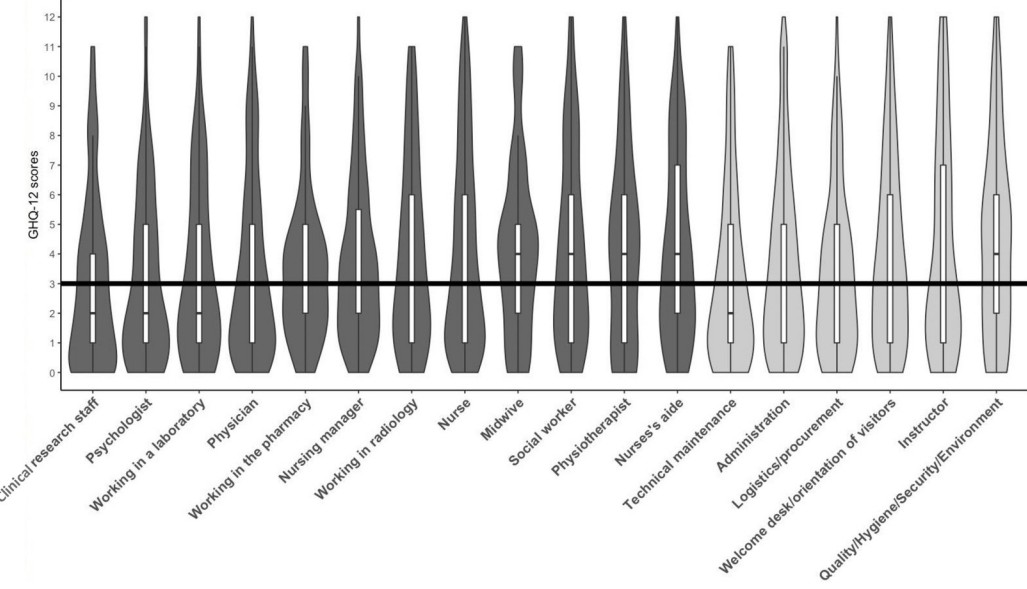

**Fig 2. Histogram and quartiles of GHQ-12 scores by profession.** The black line represents the threshold of GHQ-12 scores signifying the likely presence of psychological distress. Professions considered as "medical/caregiving" are shown in dark grey, and "non-medical" professions in light grey.

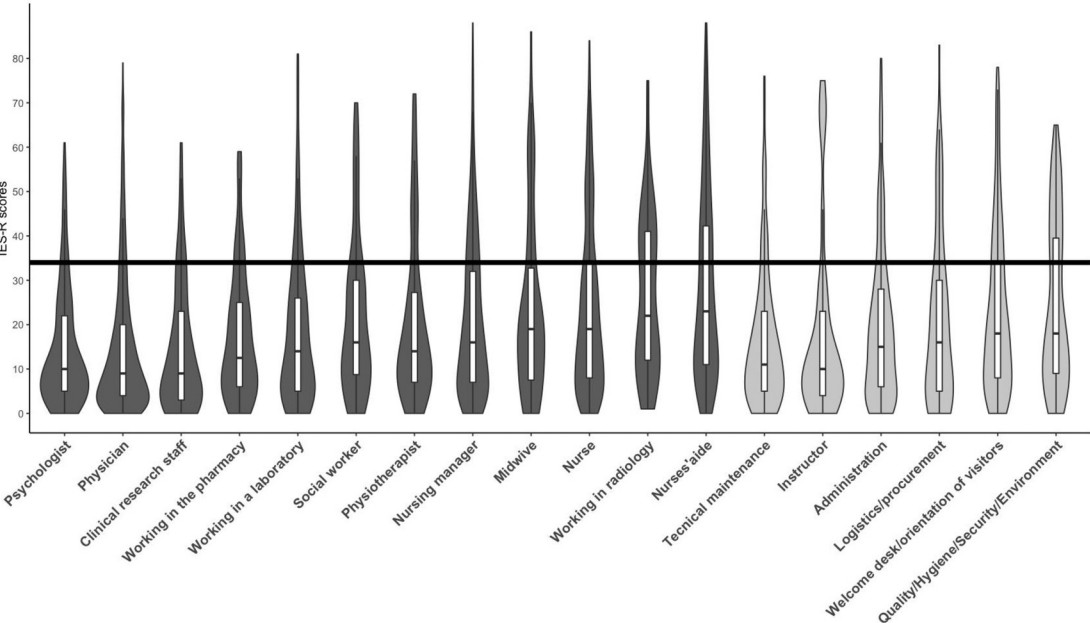

**Fig 3. Histogram and quartiles of IES-R scores according to profession.** The black line represents the threshold value of IES-R scores indicative of the possible presence of post-traumatic stress disorder at one month after the event. Medical/caregiving professions are shown in dark grey, and non-medical professions in light grey.

aggressiveness of the medical/caregiving staff in response to changing directives and the lack of PPE. Furthermore, some respondents reported being threatened with being held responsible if a caregiver was contaminated, or having received aggressive phone calls.

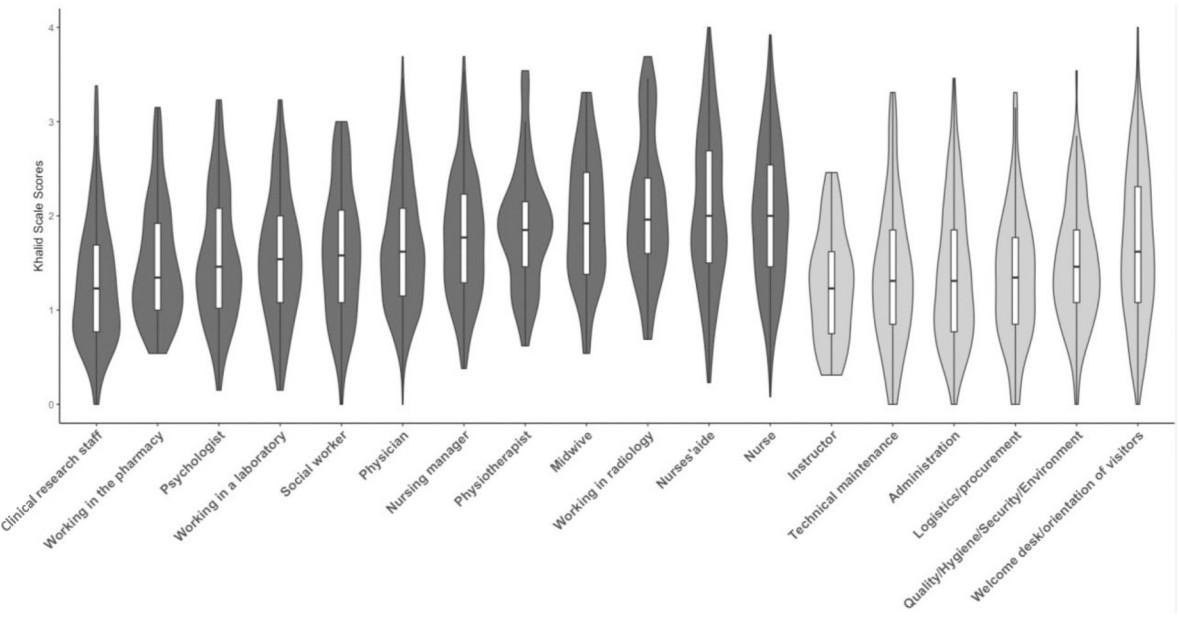

**Fig 4. Histogram and quartiles of scores on Khalid's scale, by profession.** Medical / caregiving professions are shown in dark grey, and non-medical professions in light grey.

**Table 2. Analysis of text answers to the open-ended questions identifying the three most frequently cited difficult situations for each professional group.**

| Professions | Difficult situation 1 | Difficult situation 2 | Difficult situation 3 |
|---|---|---|---|
| **Medical / Caregiving Professions** | | | |
| **Laboratory workers (n = 105)** | Workload (41%) | Lack of PPE (24.8%) | Constantly changing protocols (15.2%) |
| **Physicians (n = 698)** | Workload (28.1%) | Lack of PPE (14.2%) | Emotional management of colleagues (12.3%) |
| **Clinical research (n = 110)** | Workload (27.3%) | Working from home (19.1%) | The urgency of the situation (16.4%) |
| **Nurses' aides (n = 205)** | Lack of PPE (41.5%) | Constraints of hygiene protocols (33.2%) | Workload (24.4%) |
| **Nurses (n = 912)** | Lack of PPE (20.3%) | Risk /Fear of contaminating family (16.8%) | Changing units / hospitals (15.7%) |
| **Radiology staff (n = 69)** | Constraints of hygiene protocols (47.8%) | Lack of PPE (46.4%) | Workload (40.6%) |
| **Physiotherapists (n = 62)** | Constraints of hygiene protocols (41.9%) | Workload (25.8%) | Constantly changing protocols (17.7%) |
| **Pharmacy staff (n = 110)** | Difficulty obtaining drugs and devices (47.3%) | Workload (40%) | Difficulty obtaining PPE (28.2%) |
| **Midwives (n = 74)** | Patients' isolation from families (32.4%) | Constantly changing protocols (31.1%) | Lack of PPE (27%) |
| **Psychologists (n = 194)** | Tele-consultation (24%) | Providing support for caregivers (19.1%) | Lack of PPE (7.8%) |
| **Nursing managers (n = 396)** | Emotional management of caregivers (45.7%) | Workload (30.3%) | Managing work schedules (26.5%) |
| **Social workers (n = 78)** | Closures, difficulties contacting extramural services (52.6%) | Working from home (28.2%) | Professional isolation (23.1%) |
| **Non-medical professions** | | | |
| **Quality, hygiene, security, environment (n = 134)** | Workload (28.4%) | Lack of PPE (23.9%) | Aggressiveness of other professions towards me (17.9%) |
| **Maintenance/technical staff (n = 99)** | Workload (27.3%) | Lack of PPE (22.2%) | Lack of information (18.2%) |
| **Welcome desk / orientation (n = 221)** | Workload (20.4%) | Lack of PPE (19%) | Lack of information (17.6%) |
| **Administration (n = 390)** | Workload (26.7%) | Professional isolation (16.4%) | Emotional management of colleagues (14.1%) / Risk, fear of being contaminated (14.1%) |
| **Procurement/logistics (n = 98)** | Managing supply (41.8%) | Fear of not finding necessary equipment/material (25.5%) | Workload (24.5%) |
| **Instructors/training staff (n = 53)** | Working from home (66%) | Workload (37.7%) | Changes in methods of delivering training (20.8%) / Unable to do my job properly (20.8%) |
| **All professions (N = 4008)** | Workload (24.1%) | Lack of PPE (17.9%) | Constraints of hygiene protocols (11.2%) |

Numbers in parentheses correspond to the frequency each item was cited according to the number of participants in each professional category. PPE = personal protective equipment.

## Relation between perceived stress, coping strategies, and mental health

To manage their emotions, professionals used various coping strategies, mostly positive thinking (2.6±0.02), and least frequently, avoidance strategies (2.3±0.02), as compared to the other strategies (problem solving = 2.5±0.02; seeking social support = 2.4±0.02) (all $p < .05$).

Moderation analyses showed that being female was associated with greater psychological distress, and more severe PTSD symptoms. Living in different circumstances than usual, working part-time, and changes in the work organisation were factors associated with greater psychological distress. Being older, being single/widowed/divorced, and an increased volume of work hours during the COVID-19 crisis were factors associated with more severe symptoms of PTSD (Table 3).

**Table 3. Results of linear regression analyses for the severity of psychological distress and PTSD symptoms.**

| | b | Standard error | t | p | 95%CI | |
|---|---|---|---|---|---|---|
| | | | | | Lower | Upper |
| **Final model with GHQ-12; AIC = 16716** | | | | | | |
| Female sex | 0.1 | 0.1 | 2.3 | .023 | 0.02 | 0.2 |
| Different living conditions during crisis | 0.2 | 0.1 | 2.3 | .02 | 0.04 | 0.4 |
| Working part-time | 0.1 | 0.1 | 2.5 | .012 | 0.03 | 0.2 |
| Change in organisation of work | 0.1 | 0.04 | 3.2 | .001 | 0.05 | 0.2 |
| Perceived stress | 2.4 | 0.2 | 10.1 | < .001 | 1.9 | 2.8 |
| Coping strategy*perceived stress related to COVID-19 | | | | | | |
| Social support | -0.2 | 0.1 | -2.5 | 0.015 | -0.4 | -0.04 |
| Problem solving | 0.1 | 0.1 | 1.6 | 0.108 | -0.03 | 0.3 |
| Avoidance | -0.1 | 0.1 | 1 | 0.331 | -0.2 | 0.1 |
| Positive thinking | -0.3 | 0.1 | -5.2 | < .001 | -0.5 | -0.2 |
| **Final model with IES-R; AIC = 28185** | | | | | | |
| Female sex | 1.6 | 0.3 | 5.2 | < .001 | 1 | 2.1 |
| Age | 1.4 | 0.3 | 4.2 | < .001 | 0.7 | 2 |
| Single | 0.9 | 0.3 | 3.2 | .001 | 0.3 | 1.4 |
| Working part-time | 0.4 | 0.3 | 1.5 | .123 | 0.1 | 1 |
| Change in organisation of work | 0.4 | 0.2 | 1.8 | .08 | -0.1 | 0.9 |
| Perceived stress | 12 | 1.2 | 9.7 | < .001 | 9.6 | 14.4 |
| Coping strategy*perceived stress related to COVID-19 | | | | | | |
| Social support | -0.6 | 0.4 | -1.4 | .176 | -1.4 | 0.3 |
| Problem solving | 1.7 | 0.4 | 4.5 | < .001 | 1 | 2.4 |
| Avoidance | 0.9 | 0.4 | 2.4 | .017 | 0.2 | 1.6 |
| Positive thinking | -2 | 0.3 | -6 | < .001 | -2.7 | -1.3 |

CI = Confidence interval; AIC = Akaike Information Criterion; GHQ-12 = General Health Questionnaire; IES-R = Impact of Event Scale-Revised.

Analyses revealed a significant effect of the intensity of perceived stress on both the severity of psychological distress (B = 2.4, 95% confidence interval (CI) = 1.9, 2.8) and the severity of PTSD symptoms, (B = 12, 95%-CI = 9.6, 14.4). The positive thinking coping strategy significantly moderated the relation between perceived stress and both severity of psychological distress (B = -0.3, 95%-CI = -0.5, -0.2 and the severity of PTSD symptoms (B = -2, 95%-CI = -2.7, -1.3). The more the professionals engaged in positive thinking, the less the perceived stress, and the less severe the associated symptoms of mental health disorders. Furthermore, seeking social support (B = -0.2, 95%-CI = -0.4, -0.1] significantly moderated the relation between perceived stress and psychological distress. Conversely, the use of avoidance strategies (B = 0.9, 95%-CI = 0.2, 1.6) and problem-solving (B = 1.7, 95%-CI = 1, 2.4) potentiated the relation between perceived stress and severity of PTSD symptoms (Fig 5).

## Discussion

This study aimed to investigate the psychological impact of the COVID-19 crisis, more than one month after the peak of the first wave, among all professionals working in healthcare establishments across France. To the best of our knowledge, this is the first study to investigate such a comprehensive population in the hospital setting in France.

The study was designed to begin more than one month after the peak of the first wave of the epidemic in French hospitals (13 April 2020 according to the French public health agency),

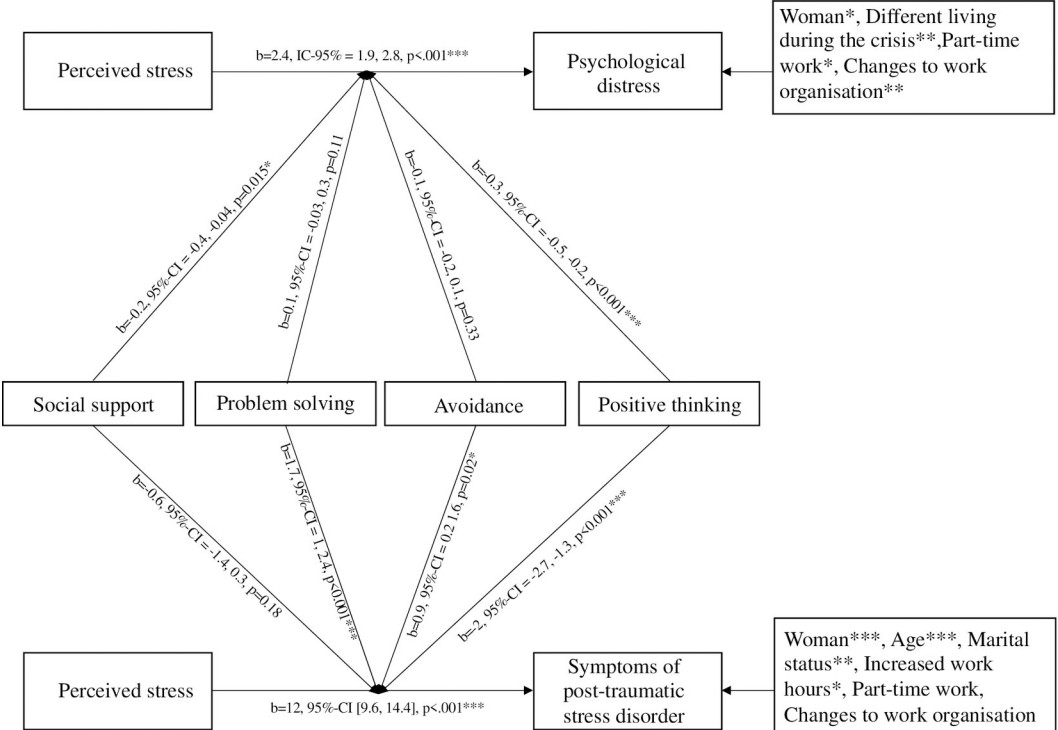

**Fig 5. Schematic representation of the moderating effect of coping stress on the relation between stress and mental health.**

while at the same time covering a period in which a rapid decline in the number of hospitalized patients was observed. This made it possible to perform the study as far as possible from the traumatic event, and to measure the negative impact in terms of mental health, and not simply the extent of acute psychological distress. The main findings are that 56.9% of all professionals were suffering from mental distress, and 21.2% had signs of PTSD at more than one month after the peak of the first COVID-19 wave. Around 20% of professionals who participated in this study had clinical symptoms of both psychological distress and psychological trauma. There was no difference between medical/caregiving staff and non-medical staff. The latter result conflicts with two European studies which found that medical professionals had fewer psychological disorders than non-medical staff [40,41], and with a third study that found the opposite to these two European studies (medical professionals > non-medical professionals) [42]. In these studies, compared to our study, professions included was less varied (medical staff: physicians, nurses; paramedics) and the non-medical people did not necessarily work in the hospital (teachers, office staff, psychologists, retired persons, social workers; unspecified), which could explain the differences in results. Regarding the frequency of mental disorders among health professionals during the COVID-19 crisis, a recent international meta-analysis found an overall prevalence of psychotraumatic disorders of 31.4% (17,5–47.3) [15]. In the general population, an international meta-analysis reported an overall prevalence of distress of 35% (23%-47%) and a prevalence of post-traumatic stress symptoms/disorders of 16% (15%-17%) [43]. Comparing our results to these studies is difficult. Indeed, one study on the impact of the COVID-19 crisis found differences between countries. Individuals in Hong Kong showed more psychological distress than those in France, for example [44]. In the same way, another study also highlights that in France and the UK, individuals (medical and non-

medical) experienced more COVID-19-related health problems than in other European countries such as Italy. The authors explain this difference by the high prevalence of COVID-19 and the high number of deaths in these two countries [40]. However, if we compare our data with those already found in France during the COVID-19 crisis, we can see that the frequencies of health disorders are relatively similar. For example, a frequency of 20.6% to 27% for PTSD symptoms was found among health professionals [45,46], and, in the general population, 35.5% for peritraumatic distress [47] and 22.3% for severe psychological distress [48].

Staff working in radiology, QSHE, and nurses' aides were those who had the highest rates of mental distress and symptoms of potential PTSD. It is surprising to note that despite a significantly lower level of perceived stress related to COVID-19 in comparison with other professionals, staff working in the QSHE sector were among those who were most markedly affected by the crisis. Based on our qualitative analysis, it would appear that the reason for this is that they were more affected by other sources of stress, very specific and not widely investigated in the literature, or not evaluated by existing measures (such as Khalid's scale). Indeed, these professionals, who are responsible for communicating and enforcing hygiene measures, reported substantial exposure to the aggressiveness of other professional groups. This aggressiveness was largely due to the difficulties professionals in other sectors faced in applying the (often contradictory and frequently changing) recommendations, the lack of PPE, and the fear of being contaminated, and/or the risk of contaminating their family. It is important to underline that all sectors mentioned the stress associated with the media coverage of the crisis (television, newspapers, social networks), as well as the uncertainty surrounding the ability to control the epidemic. The impact of information relayed through the media, a veritable "infodemic" [49] (i.e., an abundance of information including false or misleading information in digital and physical environments), has previously been cited as a risk factor for the development of mental health pathologies [50,51]. Several studies have reported that frequent exposure to social medial or information relating to COVID-19 was a source of anxiety and symptoms of stress [52], and could expose people to potentially false information or reports, or even misinformation [53], consequently amplifying existing anxiety. In the qualitative results, the workload was often cited. For the medical/caregiving staff, the workload can be explained by the prophylactic measures required to prevent or contain propagation of the virus to other patients or colleagues, such as putting on and taking PPE, and applying specific decontamination procedures. These measures, albeit necessary, are time-consuming and require additional organisation and management. In addition, some professionals saw their number of working hours double, due to colleagues being sick, or because additional beds were made available to cope with the massive influx of patients. For non-medical professionals, the search for hard-to-come-by equipment, the additional management of constantly changing safety regulations, and covering for absent colleagues undoubtedly contributed to the perceived increase in workload.

This study also made it possible to identify personal and professional factors associated with increased vulnerability to the development of mental health disorders. The severity of PTSD symptoms (assessed by the IES-R) was associated with both personal factors (e.g., being single/widowed/divorced) and professional factors (e.g., increased volume of work hours). The same was true for psychological distress (assessed by the GHQ-12), where personal conditions, such as living circumstances during the crisis, combined with professional conditions, such as part-time work and changes in work organisation, to compound the risk. These findings are similar to other reports indicating that during the pandemic crisis, some personal factors exacerbate the risk of developing mental health disorders, such as having a family member at risk of a severe form of COVID-19, living alone, or reduced social interactions [5]. Women have also been consistently reported to be at higher risk during the present crisis, with Prados

and Zamarro [54] underlining that the burden of childminding falls predominantly on women in two-parent families. The epidemic may have contributed to increasing the professional and family workload borne by women, particularly after schools were closed to contain the epidemic, and children had to do home-schooling.

Our study shows that coping strategies such as seeking social support and positive thinking can help to offset the negative effects of the crisis on psychological distress [55–58]. We also highlight the protective role of positive thinking, which makes it possible to regulate negative emotions, and transform them into more positive ones [59–61]. Positive thinking is a skill that can be improved [62] using techniques such as applications [63], cognitive-behavioural therapy online [64] or mindfulness [65]. In an epidemic context with national lockdown, it is therefore possible to propose preventive interventions to professionals working in healthcare establishments that can enhance their capacity for positive thinking. In light of our results, it would be useful to envisage management strategies that promote social support and positive thinking. It should be emphasized that coping strategies such as avoidance and problem solving, did not contribute significantly to the association between COVID-19-related stress and mental health, and may even have compounded it. In the presence of an uncontrollable public health crisis whose outcome is uncertain, it is illusory to imagine that we can avoid or solve it. Thus, these strategies (particularly problem solving) are useful and beneficial only when the situation is perceived as being amenable to change [66].

Although this is the first study, to the best of our knowledge, the investigate the impact of the COVID-19 pandemic on mental health across all professional sectors in the hospital setting in France using a mixed-methods, multicentre approach, contributing to a broader understanding of the risk factors for mental health disorders, we nonetheless acknowledge some limitations. First, the study was cross-sectional, thus precluding any conclusion of a causal relationship. Second, we do not know the level of psychological distress and symptoms of potential PTSD of the population prior to the current healthcare crisis. It is possible that the professions shown to be most affected were actually professions with a greater baseline mental health problems, for example due to a lack of recognition within their establishment. In addition, it would be relevant to assess PTSD with other recognised tools (to support our observations), such as the PCL-5 [67]. Thirdly, we could not include an exhaustive list of all types of professions working in healthcare establishment. Professions for which fewer than 50 persons responded to the questionnaire were excluded (e.g., clergy, documentalists, unions, dieticians, psychomotor therapists, ergotherapists) for reasons related to statistical power. Fourthly, this study is based on a convenience sample (leading to possible volunteer bias), and could reflect higher response rates in individuals who feel particularly concerned by suffering in the workplace, or with higher levels of work-related distress. There may thus be some over-estimation of the rates of psychological distress and potential PTSD in certain professional categories. However, in view of the observed rates of mental health disorders, it is nonetheless likely that professionals working in healthcare establishments in France were markedly affected by the pandemic. Furthermore, any putative over-estimation is offset by the fact that we included professionals most capable of describing their psychological state. Those who were too strongly affected by the crisis to be capable of responding to a questionnaire, or absent from work due to the mental health issues, were probably not captured. In addition, a document from the national association for continuing education of hospital staff (Association nationale pour la formation permanente du personnel hospitalier, ANFH) describing the distribution of professions in the hospital setting shows that our study population is congruent with the reality on the ground in terms of male to female ratio, age, ratio of medical to non-medical professions, and the ratio of full-time to part-time work [67]. Some minor differences compared to this document can be explained by the heterogeneity of healthcare establishments included (public

and private), which is a strength in terms of representativeness. The volunteer bias might also have prompted a disproportionately higher rate of participation of hospitals in geographical zones that were hardest hit by the COVID-19 epidemic. However, the description of the geographic spread of participating centres shows that they were spread across the whole country, even though there were more centres in the North and East (S1 Table). Finally, the perceived stress scale used in this study did not cover the whole spectrum of difficulties encountered by the healthcare staff, hence the importance of the open-ended questions about the participants' experience, which helped us to better understand the sources of stress.

## Conclusion

The COVID-19 pandemic has had a marked psychological impact on all professionals working in healthcare establishments in France, notably due to increased stress related to the pandemic. Staff working in radiology, and nurses' aides appear to be the professional categories most affected by the crisis, while professionals working in the QHSE sector have also been strongly affected by psychological distress and are at high risk of PTSD. This is likely due to a climate of uncertainty and the fear felt by other professionals, passed on in the form of aggressive behaviours towards QHSE professionals, who were often involved in implementing protective measures. The implementation of mental health support services for professionals, and management strategies in healthcare establishments should take into account the importance of positive thinking and social support in counterbalancing mental health disorders during an epidemic, not only during the acute phase of the crisis, but also in the longer term.

## Supporting information

**S1 Table. List of French departments participating in the study (PsyCOVID all professionals–June-September 2020, France).**
(DOCX)

**S2 Table. Number of participants in each professional category having either GHQ-12 and/or IES-R scores above the threshold indicative of the presence of the disorder (PsyCOVID all professionals–June–September 2020, France).**
(DOCX)

## Acknowledgments

The authors thank all the professionals who participated in the study, and the members of the CIC-EC1432 involved in the project, namely Emilie Galizzi, for practical coordination, and Delphine Pecqueur, for database management.

## Author Contributions

**Conceptualization:** Alicia Fournier, Alexandra Laurent, Florent Lheureux, Christine Binquet, Jean-Pierre Quenot.

**Data curation:** Alicia Fournier, Alexandra Laurent, Florent Lheureux.

**Formal analysis:** Alicia Fournier, Alexandra Laurent, Florent Lheureux.

**Funding acquisition:** Jean-Pierre Quenot.

**Methodology:** Alicia Fournier, Alexandra Laurent, Florent Lheureux, Christine Binquet, Jean-Pierre Quenot.

**Validation:** Alicia Fournier, Alexandra Laurent, Florent Lheureux, Marie Adèle Ribeiro-Marthoud, Fiona Ecarnot, Christine Binquet, Jean-Pierre Quenot.

**Writing – original draft:** Alicia Fournier, Alexandra Laurent, Florent Lheureux, Christine Binquet.

**Writing – review & editing:** Alicia Fournier, Alexandra Laurent, Florent Lheureux.

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
