## [Decision Letter · Decision Letter 0]

2 Nov 2021

PONE-D-21-21660Impact of the COVID-19 pandemic on the mental health of professionals in 77 hospitals in FrancePLOS ONE

Dear Dr. Fournier,

Thank you for submitting your manuscript to PLOS ONE. After careful consideration, we feel that it has merit but does not fully meet PLOS ONE’s publication criteria as it currently stands. Therefore, we invite you to submit a revised version of the manuscript that addresses the points raised during the review process.

We look forward to receiving your revised manuscript.

Kind regards,

Jianguo Wang, PhD

Academic Editor

PLOS ONE

“The authors thank the French Ministry for Health, who partially funded this study through the national programme for hospital research (Programme Hospitalier de Recherche Clinique National, PHRC-COVID 2019). We also thank all the professionals who participated in the study, and the members of the CIC-EC1432 involved in the project, namely Emilie Galizzi, for practical coordination, and Delphine Pecqueur, for database management.”

Reviewers' comments:

Reviewer's Responses to Questions

**Comments to the Author**

1. Is the manuscript technically sound, and do the data support the conclusions?

Reviewer #1: Yes

2. Has the statistical analysis been performed appropriately and rigorously? 

Reviewer #1: Yes

3. Have the authors made all data underlying the findings in their manuscript fully available?

Reviewer #1: Yes

4. Is the manuscript presented in an intelligible fashion and written in standard English?

Reviewer #1: Yes

5. Review Comments to the Author

Reviewer #1: This large-scale study examines the frequency of psychological distress, post-traumatic stress disorder (PTSD) and pandemic-related stress in a broad spectrum of professionals across multiple regions of France, and their relationship to demographic and work-related variables as well as to coping skills.

Though there are several individual studies addressing this question, including some from France, the current study is meritorious in view of its size and scope.

Nevertheless, there are certain minor issues that require correction on the part of the authors:

1. In the abstract, it is stated that "about two-thirds" (i.e., around 66.7%) of the sample screened positive for psychological distress; in the text, the figure given is around 57%, which is much lower. Please check this and ensure that the two statements are uniform.

2. In the Introduction, it would be helpful to refer to the multiple meta-analyses of psychological distress / depression / anxiety / PTSD in professionals (particularly healthcare professionals) during this pandemic, in addition to the individual studies cited in references 5-9.

3. The authors have used a threshold of 3 for the GHQ-12 to identify psychological distress in their sample; other researchers have sometimes used a cut-off value of 2. What was the rationale for selecting the former? Would the study's findings be substantially altered if the lower value was adopted?

4. There are several instruments besides the IES-R which have been used to screen for PTSD during the pandemic. What were the perceived advantages / merits of this instrument, according to the authors?

5. Was the modified version of the Khalid questionnaire (i.e., the one modified by the current authors as mentioned in line 159) subjected to testing or validation in a smaller sample prior to its use in the study?

6. Line 252: as the IES-R is a screening instrument, it would be more accurate to say that the subjects "screened positive for PTSD", or "scored above the cut-off for probable PTSD".

7. Lines 366-371: it would be beneficial, in addition to the general discussion that follows, to compare the current study's findings to those of others in France and adjacent European countries, both in the general population and in professionals or healthcare workers.

6. PLOS authors have the option to publish the peer review history of their article (what does this mean?). If published, this will include your full peer review and any attached files.

Reviewer #1: **Yes: **Ravi Philip Rajkumar

---

## [Author Response · Author response to Decision Letter 0]

17 Dec 2021

Dear Editor, Dear reviewer, 

We would like to thank the reviewer for his constructive comments, which helped us to improve the article. All his suggestions and remarks have been taken into account, and our point-by-point responses are described in the attached document.

The changes have been made within the text in green to make them visible. 

We thank you again for your interest in our work.

PhD Alicia Fournier

Reviewer #1: Ravi Philip Rajkumar

1. In the abstract, it is stated that "about two-thirds" (i.e., around 66.7%) of the sample screened positive for psychological distress; in the text, the figure given is around 57%, which is much lower. Please check this and ensure that the two statements are uniform.

Thank you for pointing out this error. We have made a correction in the abstract 

2. In the Introduction, it would be helpful to refer to the multiple meta-analyses of psychological distress / depression / anxiety / PTSD in professionals (particularly healthcare professionals) during this pandemic, in addition to the individual studies cited in references 5-9.

Thank you for this advice. We have added meta-analyses lines 40-46. 

3. The authors have used a threshold of 3 for the GHQ-12 to identify psychological distress in their sample; other researchers have sometimes used a cut-off value of 2. What was the rationale for selecting the former? Would the study's findings be substantially altered if the lower value was adopted?

The choice of a cut-off value of 2/3 was made following a brief review of the literature. Goldberg and colleagues (1997) [1] recommend a cut-off of 2/3. However, depending on the population studied, the cut-off may vary. Since then, other studies have been carried out on the cut-off value. Most studies use a cut-off of 2/3 [2–6]. Some use a cutt-off of 3/4 ([7], for more details see [8]).

In view of the literature, we have not made any changes to the cut-off in the paper. However, we have carried out new analyses in response to the author's comment. With a cut-off of 1/2, 71% of professionals show psychological distress (71.05% among the medical/caregiving staff vs. 70.70% among the non-medical staff, p=.848). QHSE were the most affected (79.2%), followed by midwives (79.2%), and professionals working in the pharmacy (79.1%), while psychologists (62.4%),clinical research staff (60.8%) and those working in technical maintenance/computer networks had the lowest levels (60.2%). Thus, the frequency of distress increases in all groups, but does not change the rationale of the study. For the regression analyses, we considered the linear score and not the cut-off.

1. Goldberg DP, Gater R, Sartorius N, Ustun TB, Piccinelli M, Gureje O, Rutter C, (1997) The validity of two versions of the GHQ in the WHO study of mental illness in general health care. Psychol Med 27: 191–197

2. Lundin A, Hallgren M, Theobald H, Hellgren C, Torgén M, (2016) Validity of the 12-item version of the General Health Questionnaire in detecting depression in the general population. Public Health 136: 66–74

3. Makowska Z, Merecz D, Mościcka A, Kolasa W, (2002) The validity of general health questionnaires, GHQ-12 and GHQ-28, in mental health studies of working people. Int J Occup Med Environ Health 15: 353–362

4. Ruiz-Frutos C, Delgado-García D, Ortega-Moreno M, Duclos-Bastías D, Escobar-Gómez D, García-Iglesias JJ, Gómez-Salgado J, (2021) Factors Related to Psychological Distress during the First Stage of the COVID-19 Pandemic on the Chilean Population. J Clin Med 10: 5137

5. Sun Y, Chen X, Cao M, Xiang T, Zhang J, Wang P, Dai H, (2021) Will Healthcare Workers Accept a COVID-19 Vaccine When It Becomes Available? A Cross-Sectional Study in China. Front Public Heal. doi: 10.3389/fpubh.2021.664905

6. Gelaye B, Tadesse MG, Lohsoonthorn V, Lertmeharit S, Pensuksan WC, Sanchez SE, Lemma S, Berhane Y, Vélez JC, Barbosa C, Anderade A, Williams MA, (2015) Psychometric properties and factor structure of the General Health Questionnaire as a screening tool for anxiety and depressive symptoms in a multi-national study of young adults. J Affect Disord 187: 197–202

7. Fattori A, Cantù F, Comotti A, Tombola V, Colombo E, Nava C, Bordini L, Riboldi L, Bonzini M, Brambilla P, (2021) Hospital workers mental health during the COVID-19 pandemic: methods of data collection and characteristics of study sample in a university hospital in Milan (Italy). BMC Med Res Methodol 21: 163

8. Marvaldi M, Mallet J, Dubertret C, Moro MR, Guessoum SB, (2021) Anxiety, depression, trauma-related, and sleep disorders among healthcare workers during the COVID-19 pandemic: A systematic review and meta-analysis. Neurosci Biobehav Rev 126: 252–264

4. There are several instruments besides the IES-R which have been used to screen for PTSD during the pandemic. What were the perceived advantages / merits of this instrument, according to the authors?

IES-R is one of the most common used questionnaire for assessing post-traumatic stress symptoms across different cultures, settings and types of trauma [9]. A total score of 33 on the IES-r yielded diagnostic sensitivity of 0.91 and specificity of 0.82 [10]. Although the IES-R does not fully align with the DSM-V criteria, it does align with three criteria. 

9. Weiss DS, Marmar CR. The Impact of Event Scale-Revised. In Wilson JP, Keane TM (Eds.), Assessing psychological trauma and PTSD. New York: Guilford Press;1997.

10. Creamer M, Bell R, Failla S. Psychometric properties of the impact of event scale - revised. Behav Res Ther. 2003;41(12):1489–96. https://doi.org/10.1016/j.brat.2003.07.010.

5. Was the modified version of the Khalid questionnaire (i.e., the one modified by the current authors as mentioned in line 159) subjected to testing or validation in a smaller sample prior to its use in the study?

In a previous study of ICU professionals, we showed that these items constitute a single factor [11]. However, here, we did not pre-test the questionnaire. The changes correspond to the name of the disease (MERS-CoV became COVID-19) and we deleted items that did not concern non-medical professionals (e.g., “Seeing your colleagues getting intubated”) or items that reflected the consequences of stress and not the stressors e.g., You had physical stress/fatigue). « You were emotionally exhausted » item was replaced by an item that reflected a reality in the field in France “Recommendations and protocols are evolving/changing rapidly”. 

[11] Laurent A, Fournier A, Lheureux F, Louis G, Nseir S, Jacq G, et al. Mental health and stress among ICU healthcare professionals in France according to intensity of the COVID-19 epidemic. Ann Intensive Care 2021;11:90. https://doi.org/10.1186/s13613-021-00880-y.

6. Line 252: as the IES-R is a screening instrument, it would be more accurate to say that the subjects "screened positive for PTSD", or "scored above the cut-off for probable PTSD".

Thank you for this clarification. We have made the changes (e.g., lines 9, 116, 211, 214, 321, 394, 405)

7. Lines 366-371: it would be beneficial, in addition to the general discussion that follows, to compare the current study's findings to those of others in France and adjacent European countries, both in the general population and in professionals or healthcare workers.

Thank you for the improved discussion. We have added a paragraph comparing our results to the literature (lines 324-346).

---

## [Decision Letter · Decision Letter 1]

25 Jan 2022

Impact of the COVID-19 pandemic on the mental health of professionals in 77 hospitals in France

PONE-D-21-21660R1

Dear Dr. Fournier,

We’re pleased to inform you that your manuscript has been judged scientifically suitable for publication and will be formally accepted for publication once it meets all outstanding technical requirements.

Kind regards,

Jianguo Wang, PhD

Academic Editor

PLOS ONE

Additional Editor Comments (optional):

Reviewers' comments:

Reviewer's Responses to Questions

**Comments to the Author**

1. If the authors have adequately addressed your comments raised in a previous round of review and you feel that this manuscript is now acceptable for publication, you may indicate that here to bypass the “Comments to the Author” section, enter your conflict of interest statement in the “Confidential to Editor” section, and submit your "Accept" recommendation.

Reviewer #1: All comments have been addressed

2. Is the manuscript technically sound, and do the data support the conclusions?

Reviewer #1: Yes

3. Has the statistical analysis been performed appropriately and rigorously? 

Reviewer #1: Yes

4. Have the authors made all data underlying the findings in their manuscript fully available?

Reviewer #1: Yes

5. Is the manuscript presented in an intelligible fashion and written in standard English?

Reviewer #1: Yes

6. Review Comments to the Author

Reviewer #1: The revisions made by the authors are satisfactory in my opinion. I have no further major changes or corrections to suggest.

7. PLOS authors have the option to publish the peer review history of their article (what does this mean?). If published, this will include your full peer review and any attached files.

Reviewer #1: **Yes: **Ravi Philip Rajkumar

---

## [Editor Report · Acceptance letter]

8 Feb 2022

PONE-D-21-21660R1 

Impact of the COVID-19 pandemic on the mental health of professionals in 77 hospitals in France 

Dear Dr. Fournier:

I'm pleased to inform you that your manuscript has been deemed suitable for publication in PLOS ONE. Congratulations! Your manuscript is now with our production department. 

Kind regards, 

on behalf of

Dr. Jianguo Wang 

Academic Editor

PLOS ONE